# From Churchill to Elephants: The Role of Protective Genes against Cancer

**DOI:** 10.3390/genes15010118

**Published:** 2024-01-18

**Authors:** Annalisa Gazzellone, Eugenio Sangiorgi

**Affiliations:** Sezione di Medicina Genomica, Dipartimento di Scienze della Vita e Sanità Pubblica, Università Cattolica del Sacro Cuore, 00168 Rome, Italy; annalisa.gazzellone01@icatt.it

**Keywords:** cancer protective genes, Down syndrome, Laron syndrome

## Abstract

Richard Peto’s paradox, first described in 1975 from an epidemiological perspective, established an inverse correlation between the probability of developing cancer in multicellular organisms and the number of cells. Larger animals exhibit fewer tumors compared to smaller ones, though exceptions exist. Mice are more susceptible to cancer than humans, while elephants and whales demonstrate significantly lower cancer prevalence rates than humans. How nature and evolution have addressed the issue of cancer in the animal kingdom remains largely unexplored. In the field of medicine, much attention has been devoted to cancer-predisposing genes, as they offer avenues for intervention, including blocking, downregulating, early diagnosis, and targeted treatment. Predisposing genes also tend to manifest clinically earlier and more aggressively, making them easier to identify. However, despite significant strides in modern medicine, the role of protective genes lags behind. Identifying genes with a mild predisposing effect poses a significant challenge. Consequently, comprehending the protective function conferred by genes becomes even more elusive, and their very existence is subject to questioning. While the role of variable expressivity and penetrance defects of the same variant in a family is well-documented for many hereditary cancer syndromes, attempts to delineate the function of protective/modifier alleles have been restricted to a few instances. In this review, we endeavor to elucidate the role of protective genes observed in the animal kingdom, within certain genetic syndromes that appear to act as cancer-resistant/repressor alleles. Additionally, we explore the role of protective alleles in conditions predisposing to cancer. The ultimate goal is to discern why individuals, like Winston Churchill, managed to live up to 91 years of age, despite engaging in minimal physical activity, consuming large quantities of alcohol daily, and not abstaining from smoking.

## 1. Introduction

In 1992, a famous best-selling book titled “Sharks Don’t Get Cancer” caused a sensation among the general public, by claiming that sharks are immune to cancer due to the properties of cartilage or its extract [1]. This pseudoscientific claim created significant controversy, particularly within the scientific community. The controversy reached its peak when a phase III clinical trial utilizing shark cartilage extract failed to demonstrate any significant benefits for patients in the treatment of cancer [2].

The notion of using miraculous nutraceuticals for cancer treatment goes back to the origins of medicine itself. In modern times, this pseudoscience often relies on specific biological mechanisms purported to mediate the claimed anti-cancer effects. The book “Sharks Don’t Get Cancer” was notable for initiating a false claim based on a different approach: the observation, albeit somewhat valid, that a particular species appeared somehow resistant to cancer development. This concept was not entirely new, as epidemiologist Richard Peto had observed in 1970 that larger animals, contrary to common belief, experienced fewer instances of cancer than smaller animals [3,4]. Until then, cancer had been largely viewed as a linear consequence of having many cells, with one of them eventually accumulating a critical number of mutations, leading to transformation into a cancerous cell. Peto’s paradox, as it came to be known, presented a puzzle that defied simple explanation. Various attempts were made to elucidate this phenomenon in biological terms, exploring factors such as the role of the immune system and different regulatory pathways. However, any explanation had to account for the fact that specific alleles or genes in larger animals could counteract the environmental factors influencing cancer cell proliferation. Evolution likely addressed this problem in diverse ways across different species.

The sequencing of the first human genome in 2000 ushered in a new era, allowing for the direct identification of genes or gene variants associated with specific phenotypes, first in humans, then in mice, and eventually in other living organisms. Initially, the genomic revolution primarily focused on identifying disease determinants, as most genes predisposing to cancer tended to manifest in more severe phenotypes, typically in younger individuals. These predisposing genes were relatively easy to discover, especially in cases where the phenotype ran in families, affecting younger individuals and resulting in multiple tumors within the same individual [5,6,7]. This led to the identification of many genes predisposing to cancer, which paved the way for the development of numerous targeted treatments that are now available [8]. Conversely, while one of the predisposing alleles was considered the disease allele, the other could be considered the healthy allele. However, studies concentrating on the genetic underpinnings of health lagged behind, partly due to the challenges in identifying “healthy” subjects and defining health in a quantifiable manner.

In a seminal paper, Topol described alleles in genes with a protective effect as those that, through a classical loss-of-function effect variant, could significantly influence the development of disease [9,10]. This rigorous definition proved instrumental in overcoming many of the challenges associated with discovering genetic determinants of health. Moreover, once the gene-allele combination was identified, it became possible to target the same molecule in an attempt to replicate the spontaneous phenotype observed in the healthy population. One of the most compelling examples was the discovery of loss-of-function mutations in *PCSK9*, which resulted in low levels of LDL cholesterol [11]. This discovery promptly led to the production of monoclonal antibodies against PCSK9, heralding a completely new form of treatment for hypercholesterolemia [12].

The widespread availability of sequencing analysis in families with a history of severe genetic diseases unveiled a new cohort of individuals referred to as “genetic superheroes” or “human knock-outs” [13,14]. These individuals carry heterozygous or homozygous alleles for severe genetic conditions, yet remain unaffected, likely due to the presence of alleles in different genes, possibly in the same or different pathways, which counterbalance the phenotypic effects of the disease-causing allele. Thus far, alleles with a major effect explaining the “genetic superhero” phenotype have yet to be discovered.

According to the most recent data from the World Health Organization, the estimated global lifetime risk of cancer from birth to death was approximately 20% in 2020 [15]. It is estimated that only 20% of tumors are associated with a mutation in a cancer predisposition gene, while about 75–80% of cancers are sporadic and result from a combination of multiple factors (environmental, lifestyle, or medical), with a significant role also attributed to genetic background [16].

Identifying alleles that confer protection against cancer is a much more complex endeavor. It is a common observation that many individuals, despite engaging in unhealthy lifestyles for a significant portion of their lives, somehow remain protected from the most severe consequences of cancer. Winston Churchill, known for his remarkable leadership during World War II, serves as a prominent example. Despite a documented lifestyle characterized by alcohol and cigar smoking, coupled with a lack of significant physical activity, Churchill lived to the age of 91, approximately 10 years longer than the average lifespan in the Western world nowadays [17].

While creating or retrospectively studying a cohort of individuals with characteristics similar to Churchill’s would be challenging and ethically problematic, the best approach to studying the “cancer resistance or cancer protection” phenomenon often arises from extreme examples, as previously mentioned. In this review, we will outline the various approaches taken by researchers in the cancer field to investigate the determinants of health against cancer in well-known genetic syndromes such as Down syndrome, Laron syndrome, and triplet diseases. Additionally, we will explore how evolution may have resolved Peto’s paradox in animals, devising strategies to counteract large cell populations.

## 2. Reduced Cancer Risk in Genetic Syndromic Conditions

Patients with syndromic conditions have always been a population described in numerous studies relating to the co-occurrence of other morbidities. Among these studies, attempts were made to establish whether there was an increased risk of cancer in this patient population. Over time, this higher risk has been established for numerous syndromic conditions with a known genetic basis, such as overgrowth syndromes [18], RASopathies [19], phacomatosis [20], and some microdeletions syndromes, among which is velocardiofacial syndrome [21].

Although the increased risk of tumors is a known and often quantified aspect of some genetic conditions and with a defined oncological surveillance, there is not as much data on the association between specific syndromic conditions and a lower risk of malignancies. However, this information would be very useful for understanding the possible protective effect of certain mutant alleles towards the development of cancer. Many genetic syndromes could be considered as a spontaneous genetic model where the genetic cause, besides being the cause of the specific condition, could have, as a secondary effect, a role in cancer protection and the causative gene/genes could have a protective role against cancer. The three main occurrences present in the literature are Down syndrome, Laron syndrome, and the large chapter of dynamic mutations causing brain diseases.

### 2.1. Down Syndrome

Down syndrome (DS) has been extensively studied in relation to the reduced incidence of solid tumors. From a superficial analysis, DS individuals have the same incidence of cancer as the rest of the normal population. However, a comprehensive study involving a large cohort of individuals with DS revealed a decreased risk of all major groups of malignant solid tumors, except for testicular cancer, while leukemia is more frequent, especially during the pediatric stage [22]. Particularly noteworthy was the significantly lower occurrence of lung, skin, cervical, and female breast cancers [23].

This phenomenon is remarkable, considering that individuals with DS possess several major risk factors that predispose them to cancer. These factors include hypotonia leading to reduced physical activity and obesity, accelerated organ aging, immunodeficiency, dysfunction in DNA repair systems resulting in increased DNA damage, mitotic chromosomal instability, mitochondrial alterations leading to high levels of reactive oxygen species (ROS), and elevated expressions of multiple oncogenes located on chromosome 21 [24].

Various hypotheses have been proposed to explain this lower risk of solid tumors, encompassing genetic dosage effects, the overexpression of tumor suppressor/repressor genes, disrupted metabolism, impaired neurogenesis and angiogenesis, increased apoptosis, dysregulation of the immune system, and epigenetic abnormalities.

The *ETS2* (V-Ets Avian Erythroblastosis Virus E26 Oncogene Homolog 2) gene, located on chromosome 21q22.3, regulates cell survival by controlling multiple targets such as TP53, P21, CYCLIND1, PRESENILIN1, and ICAM1 [25]. Initially categorized as a proto-oncogene linked to acute megakaryoblastic leukemia when involved in somatic balanced translocations [24], a pivotal study utilizing mouse models of DS revealed that overexpression of *ETS2* in APC^min^ mice prevented the development of intestinal tumors [25,26,27]. Despite it never being found mutated in tumor-predisposing conditions, *ETS2* appears to act more precisely as a tumor repressor rather than a suppressor gene. Notably, in the same study, different dosages of *ETS2* alleles in mice corresponded closely to the incidence of intestinal tumors in the APC^min^ background [27].

Another gene, *SOD1*, located in 21q22.11, contributes to disrupted metabolism by overexpressing superoxide dismutase 1, leading to heightened ROS levels in neurons, lymphocytes, and fibroblasts. The imbalance between the cytosolic isoform of *SOD1* (elevated) and glutathione peroxidase (normal) induces intracellular ROS accumulation, consequently causing oxidative stress, cell damage, and apoptosis [28,29].

Some studies suggest that the overall impact on tumors may relate to disrupted angiogenesis, as individuals with DS often exhibit frequent benign or indolent neoplasms, but significantly fewer aggressive tumors [30]. Among the potentially “protective genes” overexpressed in DS individuals interfering with angiogenesis, the prominent one is the *COL18A1* gene.

The *COL18A1* (Collagen Type XVIII α1 Chain) gene encodes the α chain of collagen XVIII, which, upon proteolysis, transforms into endostatin, a potent inhibitor of angiogenesis, by inhibiting the pro-angiogenic factor VEGFA (vascular endothelial growth factor A). Endostatin levels are increased in individuals with DS [31]. Reduced angiogenesis could play a significant role in halting tumor progression, albeit of lesser importance in preventing its occurrence.

Special consideration must be given to the *DYRK1A* (Dual-specificity tYrosine-phosphorylation Regulated Kinase 1A) gene, which is expressed ubiquitously and encodes a protein kinase involved in various aspects of the DS phenotype. *DYRK1A* participates in the proliferation and differentiation of neuronal progenitor cells [32]. It also plays a dual role in tumorigenesis, being overexpressed in numerous cancers. *DYRK1A* promotes tumorigenesis [33] by phosphorylating the NFAT transcription factors [34], sustaining proliferation through upregulation of the RAS/MAPK signaling pathway, resisting apoptosis via caspase 9 phosphorylation, and enhancing angiogenesis through sustained accumulation of the VEGFR2 receptor [35]. However, its pro-cancer ability is counteracted by tumor repressor properties mediated by phosphorylation of TP53 serine 15 [36], inducing cell cycle arrest in G0/G1, transcriptional suppression via the DREAM complex, and playing a central role in the double-strand break repair mechanism through RNF169 phosphorylation [37].

The seemingly contradictory aspects of the DS phenotype are mirrored by the dual role of *DYRK1A*. While mouse models exist that either overexpress [38] or lack *Dyrk1a* [39], most studies using these mice focus on the neurological aspects of DS. It would be intriguing to explore, in various cancer models, how *Dyrk1a* dosage affects cancer development and progression.

In terms of the immune system, individuals with DS exhibit increased susceptibility to respiratory infections. Some propose that DS represents a primary immunodeficiency, due to global thymic hypofunction and lymphopenia of T and B cells. Paradoxically, despite this immune dysfunction predicting an increased cancer incidence, this does not occur. Some authors suggest this could be due to an increased proportion of γδT cells responsible for tumor immunosurveillance, favoring the early elimination of cancer cells [40]. However, the gene or genes responsible for this phenotype remain unidentified.

Regarding epigenetic alterations, individuals with DS display an aberrant pattern of methylation across the entire genome, with areas of both hypermethylation and hypomethylation [40]. Recurrent and reproducible epigenetic changes on chromosome 21 are observed in various tissues and cell subtypes.

Chromosome 21 contains up to 30 miRNAs, with five of them being overexpressed and linked to the DS phenotype: let-7c, miR-99a, miR-125b2, miR-155, and miR-802 [41]. Their overexpression leads to reduced expression of proteins encoded by the targeted mRNAs. If some of these proteins were oncogenic, this would protect individuals with DS from developing tumors. The overexpression of these miRNAs may, to some extent, explain the genome-wide impact of the extra chromosome 21.

Concerning the genetic dosage effect, the overexpression of certain genes has been examined. The S100B (S100 Calcium Binding Protein β) gene encodes a calcium-binding protein secreted by glial cells that induces neural cell differentiation. Its overexpression in DS promotes neurodegeneration [42] but also exhibits a protective effect against neuroblastoma [35,36,37], correlating with the absence of neuroblastoma in children with DS.

Additionally, individuals with DS manifest a premature aging phenotype across various tissues, experiencing increased genotoxic stress and oxidative DNA damage. This effect could be attributed to the overexpression of *DYRK1A* and *ETS2*, which activate the DNA damage response in multiple cells, including stem cells, leading to a stem cell exhaustion phenotype. Mitotic instability related to the extra chromosome 21 contributes to further somatic aneuploidy and somatic mosaicism. Normally, these events would lead to increased tumor growth. However, unlike other constitutional aneuploidies such as Klinefelter and Turner syndromes, DS individuals seems to exhibit rare segmental chromosomal instability and somatic chromosomal translocations, translating into a lower susceptibility to solid tumors in both pediatric and adult populations [43]. While there is debate regarding whether patients with Down syndrome exhibit a higher or lower rate of chromosomal instability compared to those without the condition, it is established that individuals with DS have a significantly lower incidence of prostate cancer. In cases where translocations involving the *ERG*-*TMPRSS251* genes are common in the early phases of prostate cancer, it is noteworthy that such occurrences are less frequent among individuals with Down syndrome. Both of these genes are located on chromosome 21 [44].

Most studies concerning the protective effects of genes on chromosome 21 in DS have emphasized the concept that likely no single gene alone can exert a significant impact. This perspective takes into account that chromosome 21 also harbors a few genes known to have an oncogenic role or to stimulate cancer cell proliferation, growth, and metastasis [24].

Among the various genes described, *DYRK1A* plays a significant role, seemingly implicated in all major phenotypic aspects of DS. For this reason, numerous biological approaches have been initiated to modulate the kinase activity of this protein. Interestingly, this approach will help to elucidate the multifaceted roles of this protein in the various aspects of cancer biology discovered thus far (Figure 1).

### 2.2. Laron Syndrome

Laron syndrome (LS) is an autosomal recessive disorder caused by dysfunction in the growth hormone receptor. It is characterized by marked short stature, typical facial features, delayed sexual development, and obesity resulting from the inability to synthesize insulin-like growth factor I (IGF1) in response to growth hormone (GH) [45,46]. In a study conducted by Laron et al., involving a cohort of 222 patients (comprising more than half of all known patients) with congenital IGF1 deficiency (169 of which had LS), none of these individuals had a history of cancer [47]. In contrast, 9–24% of their family members had a malignancy history. This near-complete protection against cancer was observed in individuals who underwent GH treatment for a certain period, suggesting that the protective role started early during prenatal life and persisted into adulthood. Heterozygous carriers did not exhibit any dosage effect of this protective mechanism. The relationship between elevated levels of IGF1 and cancer is well established, elucidating higher cancer rates in obese individuals [48,49]. Despite being obese, Laron patients remain free from cancer. Several studies on lymphoblastoid cell lines of LS revealed significant downregulation of genes including *CYCLINA1*, *AKT3*, *SP1*, *SERPINB2*, *VESICAN*, *NPNT*, and *OR5H2*, while a few genes such as *UGT2B15*, *UGT2B17*, and *TXNIP*, involved in xenobiotics detoxification and mitochondrial redox regulation, were upregulated [50].

In vitro studies on lymphoblastoid cells from LS patients indicated reduced proliferation, altered cell cycle dynamics, decreased motility, increased apoptosis, resistance to oxidative stress challenges, and elevated expression of tumor suppressor genes. Notably, significantly reduced total and phosphorylated levels of IGF1R, a gene commonly overexpressed in various cancer types, were found in LS cells [50]. This reduction correlated with parallel decreases in the phosphorylation of downstream signaling molecules AKT and ERK, which are typical mediators in the IGF1 and insulin pathways. The reduction in expression and activation of components within the IGF1R signaling axis might underlie the decrease in the mitogenic potential of LS cells.

Moreover, the distinct representation of IGF-binding proteins (IGFBPs) in LS-derived lymphoblasts could elucidate the lower tumor incidence in these patients. Specifically, mRNA levels of *IGFBP2*, *IGFBP5*, and *IGFBP6* were decreased in LS lymphoblastoids compared to healthy controls, while *IGFBP3* mRNA levels were increased in LS cells. *IGFBP3* has been recognized as an anti-oncogene in various tumors, consistent with its increased levels in LS. *IGFBP2* is typically associated with promoting tumorigenesis and T-cell proliferation, while *IGFBP5* and *IGFBP6* promote T-cell migration and act as a chemotactic agent for T-cells. Therefore, the reductions observed in these IGFBPs align with the protective role against cancer [51].

The protective effect against cancer associated with a non-functioning insulin/IGF1 pathway was confirmed in *Caenorhabditis elegans*. Mutations in the insulin/IGF-I like signaling pathway primarily increase lifespan. When this allele was combined with a null allele for a tumor suppressor gene causing germline tumors, a complete absence of tumor development was observed, suggesting that the lack of IGF1 effectively conferred resistance to tumors [52]. Studies on mice treated with growth hormone receptor antagonists have demonstrated a lower incidence of carcinogenesis [53].

An animal model of LS, the ‘Laron’ mouse, was created by disrupting the Gh receptor gene (*Ghr1*). Similar to humans with LS, Laron mice exhibit low Igf1 and elevated Gh levels. Homozygous *Ghr1* knockout mice displayed the severe phenotype typical of Laron syndrome. However, heterozygous mice for the *Ghr1* axis showed minimal growth impairment, but presented an intermediate biochemical phenotype, with decreased Ghr- and Gh-binding protein expression and slightly reduced Igf1 levels [54]. Transgenic mice expressing human GH and/or an agonist of the Igf1 receptor showed an increased incidence of breast tumor development [55]. Conversely, mice carrying a transgene causing basal breast cancer in mice, when crossed with a tamoxifen-inducible conditional KO allele for the *Gh1r* receptor, demonstrated that the ablation of the *Gh1r* axis, after tumors reached a large size, completely inhibited the growth of breast cancer cells. These findings support the idea that growth hormone receptor antagonists may reduce the growth of cancer cells [56].

Further investigations on the *Igf1r* conditional KO mice aimed to elucidate the influential effects on cancer metastatic growth. Metastatic spread is a crucial aspect of cancer mortality. In one study, cells from Lewis lung carcinoma cells (LLC) were transplanted into mice with a *Igf1r* KO background, meaning that the Igf1r axis was intact in cancer cells, but it was disrupted in the tumor microenvironment. The LLC, when transplanted into wild-type mice, led to multiple pulmonary metastases, while *Igf1r* KO mice showed a reduced tumor burden. The same effect was replicated with a subcutaneous injection of a B16 melanoma cell line [57]. These experiments in mice effectively demonstrated the potent inhibition of the Gh1r/Igf1 pathway, whether in cancer cells or in the supporting microenvironmental cells, leading to the inhibition of cancer cell growth, reduced inflammatory infiltration, enhanced apoptosis, reduced proliferation, and metastatic arrest. Ongoing trials with Gh1r/Igf1r antagonists will reveal their potential in this therapeutic approach, initiated from the clinical observation that LS patients do not develop cancer (Figure 2).

## 3. Reduced Cancer Risk in Syndromes with Dynamic Mutations

Fragile X syndrome (FXS) is caused by the absence in the brain of the fragile X mental retardation protein (FMRP). This protein is ubiquitously expressed suggesting that, in addition to its effects in brain, it may have fundamental roles in other organs [58].

From the most comprehensive cumulative report of tumors in FXS obtained from 1988 to 2013, the rate of the most common malignancies in men (prostate, lung, and colorectal) were 6.8%, 4.5%, and 2.3%, respectively. That is significantly less than the expected estimated cancer incidence by site in the normal population (21%, 14%, and 8%, respectively). The male to female ratio was 1.58 to 1% and the female incidence of the three most common cancers (breast, lung, and colorectal) were 6.8%, 4.5%, and 2.3%, respectively, in comparison with the normal population (29%, 13%, and 8%) [59,60].

There is evidence that FMRP expression can be linked to cancer. FMRP, as well as *FMR1* mRNA levels, correlate with prognostic indicators of aggressive breast cancer and lung metastasis [61]. In particular, FMRP overexpression in murine breast primary tumors enhances lung metastasis, while its reduction has the opposite effect, regulating cell spreading and invasion. FMRP binds mRNAs involved in epithelial mesenchymal transition (EMT), often a prerequisite for metastases formation and invasion, including E-cadherin and Vimentin mRNAs, which are hallmarks of EMT and cancer progression [61].

There is a statistically significant downregulation of the *WNT7A* gene in FXS patients compared to healthy subjects; for this reason, the role of the WNT7A protein has also been extensively studied. Real-time PCR in FXS patients showed a real reduction in signal intensity in FXS males compared to healthy males [62]. The reduced expression of the *WNT7A* gene and its consequent downregulation of the β-catenin pathway may be related to a potential protection of FXS patients from cancer. Some of the target genes of this pathway, including the MYC, JUN, CYCLIND, and PPARδ genes, showed moderately reduced expression in FXS patients compared to normal subjects [62].

Huntington’s disease (HD) is a progressive brain disorder caused by the pathological CAG expansion sequence in the *HTT* gene. Patients with Huntington’s disease exhibit a significantly lower cancer incidence, up to 80 percent less than the general population [63]. The analysis of data from 6540 subjects in the European Huntington’s disease network REGISTRY revealed a notably reduced age-standardized incidence rate, particularly evident in prostate and colorectal cancers, which exhibited the lowest rates [63]. Several potential factors may account for these lower cancer rates. Factors such as lower life expectancy might contribute, and there could be instances where cancer is underdiagnosed among individuals with HD, especially in later stages of the illness. This could occur as relevant signs or symptoms may be overlooked or overshadowed by HD symptoms, such as cachexia.

Turner et al. [64] conducted a study examining hospital admission records in England, revealing an increased rate of cancer diagnoses within the first year after admission for HD. Interestingly, they observed an overall decrease in the rate of cancers among HD patients, particularly pronounced when excluding the first year. This suggests that under-diagnosis might be less probable in this population, contrary to what might be expected. McNulty et al. also indicated a similar effect, reinforcing this perspective.

Notably, lung cancer was the only cancer found to occur as frequently in the HD population as in the general population, as reported by Turner et al. [64]. They attributed this finding to the higher rate of smoking among individuals with HD, a trend observed in various other psychotic conditions as well [65].

While the incidence of cancer is lower in HD patients than in age-matched controls, HD-causing CAG expansions of *HTT* accelerate the progression of breast tumors and the development of metastases in mouse models of breast cancer. In particular, Thion et al. [66] showed that the length of *HTT* CAG correlates with a lower incidence of ovarian cancer in carriers of the *BRCA2* mutation and that CAG repeat length in the long *HTT* allele can be a factor in metastasis in sporadic breast cancer (HER+ subtype) [66]. One of the possible mechanisms linking HD and cancer protection may be RNA toxicity. Murmann and colleagues constructed small interfering RNAs based on *HTT* CAG repeats; these siRNAs induce cell death in vitro in all tested cancer cell lines and slow down tumor growth in a preclinical mouse model of ovarian cancer, with no signs of toxicity to the mice [67]. To indirectly confirm the role of RNA toxicity, in other neurodegenerative conditions with polyglutamine expansions a lower cancer incidence was also observed (Figure 3).

## 4. Hereditary Cancer Syndromes

A hereditary cancer syndrome is a genetic predisposition to certain types of cancer, often with onset at an early age, caused by inherited pathogenic variants in one or more genes. The most common hereditary cancer syndromes include hereditary breast and ovarian cancer syndrome, Lynch syndrome, Li–Fraumeni syndrome, Cowden syndrome, Peutz–Jeghers syndrome, and hereditary diffuse gastric cancer. The genetic causes of most hereditary cancer syndromes have already been identified and well known [8].

However, determining the precise impact of factors designated as risk factors on increasing tumorigenesis risk, particularly in individuals with hereditary forms of cancer, is often challenging. While certain factors like age, weight, tobacco smoking, and chronic inflammation are well-known and measurable, others remain less understood, such as exposure to environmental toxic substances or dietary components. Pinpointing the exact influence of these risk factors is difficult, and identifying protective factors poses an even greater challenge. Nevertheless, the widespread use of comprehensive genetic tests has been instrumental in identifying genetic variables that can potentially alter tumor progression and confer a protective role.

Below are some studies that have tried to understand whether, in a population at increased risk of cancer, there are factors capable of modifying the clinical prognosis.

### Hereditary Breast and Ovarian Cancer Syndrome

Hereditary breast and ovarian cancer syndrome is a genetic condition that significantly increases the likelihood of developing breast, ovarian, and other cancers. Studies have indicated the involvement of β-adrenoceptors in tumor progression by regulating the immune system. Specifically, the gene *ADRB2*, encoding β2-AR, a member of the G protein-coupled receptor superfamily, has been identified as a potential protective gene in BRCA patients [68]. It appears to play a crucial role in modulating immune responses. The expression level of *ADRB2* showed a positive correlation with immune cell infiltration in BRCA cancers. Notably, its specific expression in T cell subtypes and lower expression of *ADRB2* often resulted in a poorer prognosis among BRCA patients [68].

The human DNA repair protein RAD52 (hRAD52) plays a critical role in various aspects of genome maintenance. One of its well-defined roles is as a key mediator of DNA double-strand break (DSB) repair through single-strand annealing (SSA).

The *RAD52* p.Ser346Ter allele has been associated with a reduced risk of developing breast cancer in *BRCA2* carriers, and to a lesser extent in *BRCA1* carriers. Intriguingly, the expression of *RAD52* p.Ser346Ter also diminished the stimulation of SSA observed upon *BRCA2* depletion. This demonstrates the reciprocal roles of RAD52 and BRCA2 in regulating DSB repair via SSA. Immunofluorescence analysis revealed limited nuclear localization of the mutant protein compared to the wild-type, indicating that the reduced nuclear levels of *RAD52* p.Ser346Ter might account for the impaired DSB repair through SSA. These findings suggest that deficiencies in RAD52-dependent DSB repair are associated with a decreased risk of tumors in BRCA2-mutation carriers [69,70].

Numerous GWAS (Genome Wide Association Studies) studies have presented contradictory results regarding the protective role of SNPs in prevalent cancer types and cancer predisposition syndromes. The identification of major SNPs predisposing certain effects is influenced by confounding factors, which likely impact the identification of protective alleles as well. Additionally, only a limited number of well-defined loss-of-function variants, as outlined by Topol, appear to mediate a comparable effect.

## 5. Reduced Cancer Risk throughout the Animal Kingdom

Apart from various animal models, the investigations into the cancer-protective effect mediated by genes in the animal kingdom have been explored starting from the observation that some very large animals, such as elephants and whales, with a very long lifespan, present a very low rate of cancer. However, this correlation is not perfect within the same species, such as in dogs, in which larger breeds are generally more prone to cancer development than smaller breeds. Conversely, in species of very small size, such as bats, they tend to be more cancer-resistant than mice, despite having similar sizes. Another interesting species in this respect is the naked mole-rat, which, in spite being a rodent of the size of a mouse, has an extremely long lifespan of up to 30–40 years and is extremely cancer-resistant. These studies aim to determine whether different species, particularly those with longer lifespans than humans, can provide valuable insights into the functioning of the major pathways involved in tumorigenesis, and how evolution solved the Peto’s paradox in those species.

Among the extensively studied animal models, elephants have garnered significant attention. This species exhibits an estimated cancer mortality of 4.81%, which is nearly half the observed mortality rate in humans, which ranges from 11% to 25% [71]. This disparity has been associated with elephants possessing 20 copies of the *Tp53* gene, which differ in length and sequence content [72]. Notably, in the *Loxodonta africana* species, recent research has demonstrated that the 20 isoforms of the *Tp53* gene feature distinct BOX-I Mdm2-binding motifs [73].

MDM2, an oncoprotein, primarily interacts with TP53, facilitating its degradation. This process ensures TP53 activation only when necessary or in response to cellular damage. The proposed mechanism behind the lower incidence of tumors in elephants revolves around the existence of diverse binding epitopes that interact with Mdm2. These distinct pools of Tp53 proteins result in an enhanced apoptotic response to DNA damage [73]. The variations in BOX-I sequences of Mdm2 hinder the interaction between Tp53 and Mdm2, thereby preventing Mdm2-mediated degradation of Tp53. As a result, Tp53 continues to fulfill its role as the guardian of the genome. Having multiple finely tuned copies of *Tp53* prevents cells from attempting to repair irreparable DNA damage, thereby shifting the balance towards apoptosis. This multiple-copy configuration serves as an evolutionary fail-safe mechanism against the loss of a single gene copy, as can occur in humans.

Another species extensively scrutinized for low tumor incidence is the Greenland whale (*Balaena mysticetus*). This species boasts a remarkable lifespan exceeding 200 years and displays a lower tumor incidence despite its size, which entails a larger number of cells and significantly greater replication and repair mechanisms compared to humans.

In this long-lived species, the heightened expression of certain proteins involved in oncogenesis, such as Ercc1 and Pcna, has been observed, suggesting a potential protective role against tumors. Ercc1, an endonuclease involved in DNA damage repair, specifically addresses cross-links and mediates nucleotide excision repair [74]. Activating mutations in the *Ercc1* gene have been identified in *B. mysticetus* [75], while in mice, inactivating mutations in this gene are associated with a higher incidence of tumors [76].

Furthermore, an elevated expression of the *Pcna* gene has been observed in Greenland whales [75]. This gene encodes a protein that plays a crucial role in supporting polymerase activity during DNA replication and repair processes [76].

The Pcna protein holds significant importance in maintaining the polymerase’s attachment to DNA, thus playing a fundamental role in DNA replication. In instances of DNA damage, this protein undergoes ubiquitination, triggering two DNA repair pathways: homologous recombination and nucleotide excision repair. The suggested mechanism behind the greater resistance to DNA damage and the subsequent lower incidence of tumors in Greenland whales involves the overexpression of the *Pcna* gene and, consequently, an increased production of the protein.

Similar patterns have been observed in other animal species where the expression levels of *Pcna* in specific organs show a close correlation with cell proliferation. For instance, in the livers of rats, lower levels of the Pcna protein have been linked to decreased levels of cell regeneration [77].

The naked mole-rat serves as a fascinating example for several reasons [78]. Firstly, it is a very small animal with an extraordinarily long lifespan for its size. Secondly, its longevity likely derives also from an incredible resistance to cancer, mediated by a unique mechanism. While elephants and whales exploit some well-known genes for their cancer-resistant traits, *Tp53* and *Ercc1*, both directly involved in DNA repair, apoptosis, and essential cell-cycle decisions, the scientific perspective shifts significantly when examining the naked mole-rat.

The molecule responsible for this unique resistance is a high molecular weight hyaluronan, a glycosaminoglycan (the primary non-protein molecule in the extracellular matrix), secreted by fibroblasts and produced abundantly by hyaluronan synthase 2 (Has2) during postnatal life [79]. The Has2 protein found in the naked mole-rat differs from the mouse and human sequences, due to the substitution of two highly conserved asparagines with two serines. These alterations occur within the catalytic region, resulting in the enzyme from the naked mole-rat being highly processive and producing very high molecular weight hyaluronan.

Fibroblasts from naked mole-rats exhibit early contact inhibition, halting their growth at significantly lower densities. This mechanism is likely mediated by hyaluronan within the extracellular matrix, which directly interacts with Cd44 on the cell surface. This interaction induces early contact inhibition through p16^INK4a^, resulting in reduced cell proliferation, reduced hyperplasia and metastatic potential, and, on a systemic level, reduced inflammation. Additionally, it acts as an antioxidant, decreasing damage from reactive oxygen species to DNA and proteins, and makes cells more prone to apoptosis following the loss of tumor-suppressor genes [80,81].

The final confirmation that hyaluronan mediated these effects was achieved by increasing the lifespan of mice and enhancing their cancer resistance beyond that of control mice after expressing the naked mole-rat version of Hsa2. The increased expression of hyaluronan not only extended lifespan but also ensured a healthier one in mice [82] (Figure 4).

## 6. Conclusions

Protective mechanisms against cancer mediated by genes are consistently found throughout the animal kingdom, while the impact of environmental factors such as diet on this process is generally minimal across species, except for humans.

While many studies investigating the role of oncogenes and tumor suppressor genes have utilized the genetic tools available in mice, mice, due to their small size and shorter lifespans, are highly susceptible to cancer. In optimal laboratory conditions, most mice succumb to a lymphoproliferative disorder. It is widely acknowledged that mouse fibroblasts do not undergo replicative senescence; they require two mutations (Tp53 or Rb1 and activating Hras) to transform into cancerous cells, compared to human cells, which require at least five hits [83]. Additionally, mouse telomerase remains active in most somatic tissues. Their short lifespan and predisposition to cancer mean that any genetic intervention demonstrating the protective role of genes during tumorigenesis only showcases a proof-of-concept phenotype.

Consequently, the primary insight into the role of cancer-protective genes has emerged from analyzing specific cases of human genetic syndromes. For instance, individuals with DS display resistance to the most common epithelial cancers. However, among the approximately 230 human genes, pinpointing the singular gene responsible or understanding if several genes collectively contribute to this protection poses a challenge. This complexity becomes clearer in monogenic conditions like LS, FXS, or HD, where a particular protein or mechanism is actively involved.

Future studies that harness these mechanisms in a therapeutic context will confirm the validity of these assumptions. Furthermore, examining the role of evolution in safeguarding animals of various sizes from cancer, while also prolonging their lifespans, offers novel insights. However, replicating this intricate process in humans proves considerably more challenging. Nevertheless, understanding this evolutionary selection could provide new avenues for combating one of the leading causes of death and suffering in humans, offering potential breakthroughs in the field of cancer research.

## Figures and Tables

**Figure 1 genes-15-00118-f001:**
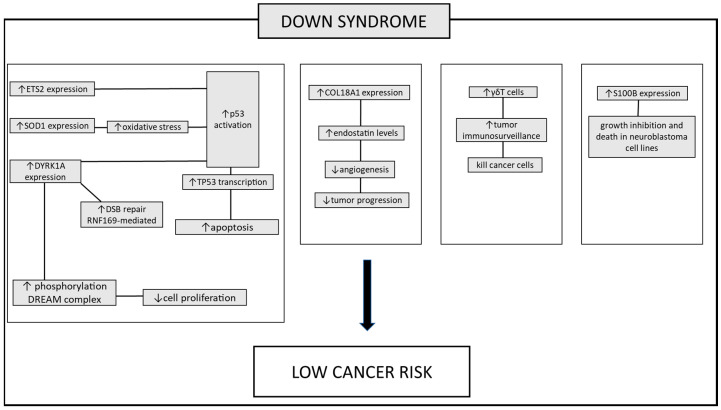
The pathways involved in protecting individuals with DS against tumors. Several mechanisms contributing to the lower incidence of tumors in these patients have been proposed. Increased TP53 apoptotic response: this is mediated by heightened expression of genes located on chromosome 21. These genes boost the transcription of *TP53*, *ETS2*, *SOD1*, and *DYRK1A*. Enhanced activation of DNA damage repair mechanisms leads to an increase in the repair of double-strand breaks, facilitated by elevated *DYRK1A* expression. The overexpression of the *DYRK1A* gene in DS patients leads to increased activation of the DREAM complex. This complex modulates the expression of genes involved in regulating the cell cycle, thereby reducing cell proliferation. Inhibition of tumor growth-promoting mechanisms: angiogenesis is suppressed due to heightened expression of *COL18A1*. Increased immune cell expression: the increased circulation of γδT cells, responsible for recognizing and suppressing tumor cells, plays a role in the lower tumor incidence. Additionally, the S100B protein, while contributing to the neurodegenerative process in DS patients, exhibits a contrasting effect in both human and murine neuroblastoma cell lines blocking tumor progression. Up and down arrows stand for “increased” or “decreased”.

**Figure 2 genes-15-00118-f002:**
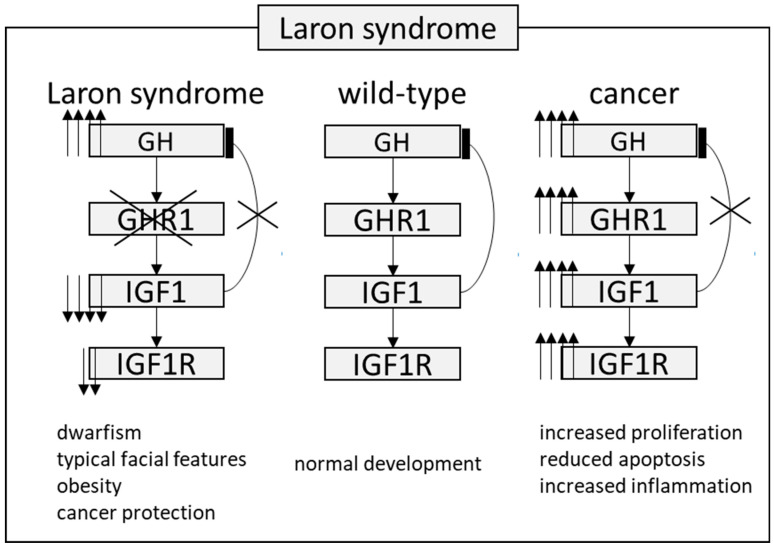
The pathway involved in protection against tumor in patients with Laron syndrome. During normal development, growth hormone (GH) is typically secreted by the pituitary gland and exerts its effects through the growth hormone receptor (GHR) present in various organs and cell types. Insulin-like growth factor 1 (IGF1) is released into the bloodstream, reaching different target organs and tissues. IGF1 negatively regulates GH secretion. However, in Laron syndrome, mutations in the GHR lead to increased GH levels, while IGF1 levels remain very low. This particular combination mediates a protective effect against cancer, despite these patients being obese. In cancer cases, a coordinated disruption of this axis at various levels results in markedly elevated IGF1 levels, thereby increasing downstream signaling. Up and down arrows stand for “increased” or “decreased”.

**Figure 3 genes-15-00118-f003:**
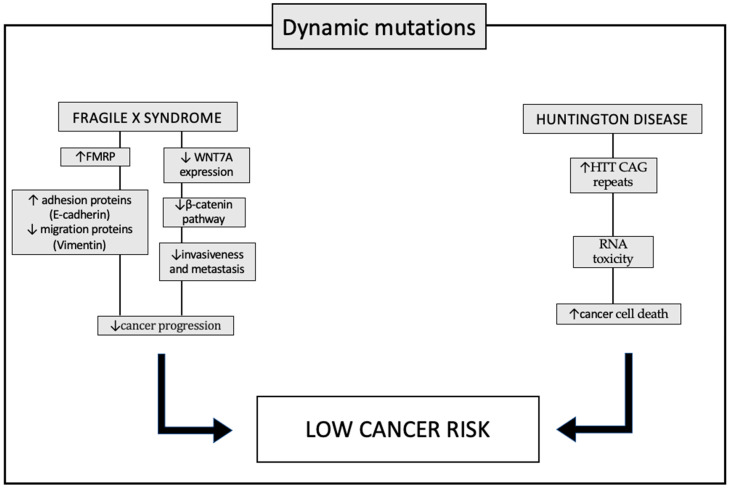
Pathways involved in protection against tumors in patient with dynamic mutations: fragile X syndrome (FXS) and Huntington’s disease (HD). In patients with FXS, increased levels of the FMRP (fragile X mental retardation 1 protein) appear to alter the expression of proteins that promote tumor progression. In particular, the typical invasiveness of tumor cells is inhibited through the decreased expression of the vimentin protein, and cell adhesion is instead promoted, thanks to the increased expression of E-cadherin. Another mechanism underlying the lower incidence of cancer in patients with FXS is the decreased expression of the WNT7A (WNT family member 7A) gene which, through downregulation of the β-catenin pathway, prevents the invasiveness of tumor cells. In patients with HD, however, one of the molecular causes supposed at the basis of the lower incidence of tumors is the increased level of RNAs, which have a toxic action, not only towards neuronal cells, causing the typical clinical signs of the disease, but also to tumor cells. Up and down arrows stand for “increased” or “decreased”.

**Figure 4 genes-15-00118-f004:**
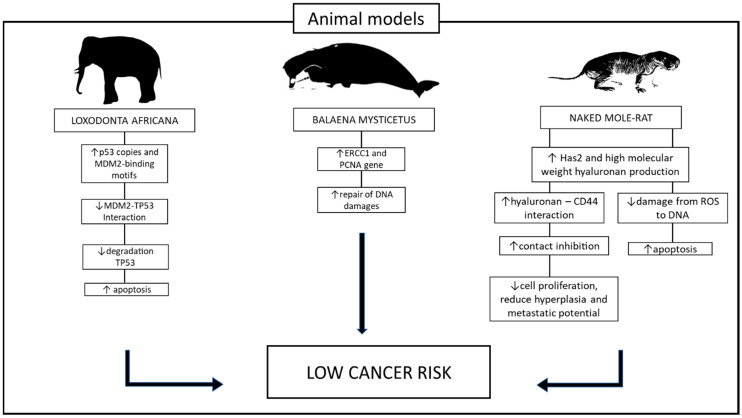
Different pathways play a role in protecting animals against tumors. In elephants, the gene *Tp53* undergoes amplification, increasing up to 20 copies. The finely regulated multiple copies of this gene, activated by DNA damage, prevent the transformation of cells with significant DNA damage into cancerous cells by inducing apoptosis. Similarly, in whales, a comparable effect is achieved through the overexpression of the ERCC1 protein. This overexpression enhances the whales’ capability to repair DNA, contributing to a heightened defense against cancer. Conversely, the naked mole-rat utilizes a completely distinct mechanism for tumor protection. These rodents synthesize an exceptionally large hyaluronan molecule, which induces early contact inhibition in fibroblasts. This process leads to reduced cell proliferation, diminished hyperplasia, decreased inflammation, and a lowered potential for metastasis. Up and down arrows stand for “increased” or “decreased”.

## Data Availability

Not applicable.

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
