# Peer review of "From Churchill to Elephants: The Role of Protective Genes against Cancer"

_genes, 2024, doi:10.3390/genes15010118_

Round 1

Reviewer 1 Report

Comments and Suggestions for Authors

This is a fascinating review of the different mechanisms that nature has developed to overcome Peto's paradox. The article examines evidence from animals and genetic syndromes, highlighting important mechanisms and key genes that are relevant to protecting against the development of cancer.

I only have the following recommendations:

It is necessary to improve the references of the following paragraph so that they accurately reflect the assertion that has been made: “This led to the identification of many genes predisposing to cancer, which paved the way for the development of numerous targeted treatments now available [8–10].” the references are related only to the inhibition of PARP in BRCA mutated tumors.

Fig 1. P53 activation instead of P53 attivation 

Author Response

We added a more appropriate reference (#8)

We made the change

Reviewer 2 Report

Comments and Suggestions for Authors

Manuscript:  From Churchill to Elephants: The Role of Protective Genes Against Cancer.

When genetic aspects (hereditary or not) that cause cancer are described, genes whose variants result in an increased risk of developing cancer are generally mentioned. However, little is known about genes whose variants are protective in the development of cancer. This is because the people who are studied are those who come to the consultation (affected by cancer), while healthy people do not consult, so they are not studied. Therefore, in the literature, data on genetic variants is largely skewed in favor of pathogenic variants, when surely there are also many protective variants to be known. This is why studies of protective variants are necessary, as well as reviews such as the present one, which focus on genes whose variants or increased copy number may be protective.

Like the authors, I think that the knowledge of genetic differences (variants, copies, gene regulation) in these gigantic organisms or in animals presenting a long longevity, can help us in the knowledge of protective mechanisms of cancer, and through this knowledge, we can think about possible therapies.

The review is well written, however, some points should be improved or better specified.

1.- As well as the probability of developing cancer according to the number of cells among DIFFERENT species is described, I think that the probability of developing cancer according to the number of cells, WITHIN the same species, should also be mentioned (e.g., larger dogs are more likely to develop cancer).

2.- Authors mention "dominant alleles / recessive alleles. Depending on the combination of alleles, the disease can be dominant or recessive (e.g. breast and ovarian cancer vs. Fanconi's anemia, both due to BRCA2 pathogenic variants). The same variant/allele can be involved in a dominant and in a recessive disease. The variant, by itself, is not dominant or recessive. It depends on the combination. Diseases, not variants, are dominant or recessive.

3.- In the Conclusions, authors write: evolutionary adaptations could provide….

I think it would be better to use the word selection instead of adaptation. The word adaptation refers to the individual. The word selection refers to the collective of a species. Natural selection is the mechanism that increases the probability of advantageous traits in the coming generations. Adaptation is the characteristic that changes according to the environment.

Author Response

1.- As well as the probability of developing cancer according to the number of cells among DIFFERENT species is described, I think that the probability of developing cancer according to the number of cells, WITHIN the same species, should also be mentioned (e.g., larger dogs are more likely to develop cancer).

> We added a sentence highlighted in yellow

2.- Authors mention "dominant alleles / recessive alleles. Depending on the combination of alleles, the disease can be dominant or recessive (e.g. breast and ovarian cancer vs. Fanconi's anemia, both due to BRCA2 pathogenic variants). The same variant/allele can be involved in a dominant and in a recessive disease. The variant, by itself, is not dominant or recessive. It depends on the combination. Diseases, not variants, are dominant or recessive.

> We replaced dominant and recessive with heterozygous and homozygous, we highlighted the changes in yellow

3.- In the Conclusions, authors write: evolutionary adaptations could provide….

I think it would be better to use the word selection instead of adaptation. The word adaptation refers to the individual. The word selection refers to the collective of a species. Natural selection is the mechanism that increases the probability of advantageous traits in the coming generations. Adaptation is the characteristic that changes according to the environment.

> We changed the world adaptations with selection, highlighted in yellow

Reviewer 3 Report

Comments and Suggestions for Authors

I very much enjoyed reading this manuscript, which really gives an interesting overview on genes, cancer, mutation across species. There are just a few comments, recommendations from my side:

1. As in the text also effect of genetic/monogenetic disease s is discussed, this should also be mentioned in the abstract

2. The authors might briefly mention the hypothesis, that the total number of tissue stem cell dicisions might correlate with cancer (as initially proposed by Tomasetti and Vogelstein (10.1126/science.1260825).

Minor points, spelling:

Figure 1: tipical à typical

Just for the sake of fanciness: Figure 2: Maybe the typesetting team could provide some nicer silhouettes for the Greenland shark and the naked mole rat

Author Response

  1. As in the text also effect of genetic/monogenetic disease s is discussed, this should also be mentioned in the abstract

> We added a sentence in the abstract

  1. The authors might briefly mention the hypothesis, that the total number of tissue stem cell dicisions might correlate with cancer (as initially proposed by Tomasetti and Vogelstein (10.1126/science.1260825).

> We could not find an appropriate place where to mention this concept without having to do a long detour about the mechanisms involved

Minor points, spelling:

Figure 1: tipical à typical

> Correction done

Just for the sake of fanciness: Figure 2: Maybe the typesetting team could provide some nicer silhouettes for the Greenland shark and the naked mole rat

> We added modified silhouette of those animals